# A conformational benchmark for optical property prediction with solvent-aware graph neural networks
Denis Potapov [1,2,7] ✉, Sergei Rogovoi[1,3,7], Kuzma Khrabrov[1], Konstantin Ushenin[1,4], Alexey Korovin[1], Anton Ber[1], Artur Kadurin [1,4,5,6] & Artem Tsypin [1]

Accurately predicting optical spectra of molecules is essential for creating better OLED emitters, solar-cell dyes, and fluorescent probes. Traditional methods, such as time-dependent density-functional theory, are computationally expensive and often inaccurate. Current Graph Neural Network (GNN) approaches for optical properties prediction are faster and offer better performance. Still, they operate on 2D graphs and ignore the 3D geometrical features that control excited-state behavior. We present nablaColors-3D, a rigorously curated dataset for the prediction of optical properties consisting of 26369 chromophore-solvent pairs with three conformations optimized at different levels of quantum theory. Based on this dataset, we establish a scaffold-split benchmark for 3D GNNs and systematically quantify how the fidelity of geometry optimization affects accuracy. Furthermore, we propose a solvent-aware modification for pretrained SE(3)-invariant architectures. Our best model, built on UniMol+, achieves MAE of 15.97 nm on a held-out test set, improving the previous state of the art by more than 30%.

Accurate prediction of molecular optical properties is fundamental to advancing research across numerous scientific and technological domains. This predictive capability serves critical needs in organic light-emitting diodes (OLEDs)[1–4], organic solar cells[5–7], biosensors and bioimaging[8–11], organic dyes and pigments[12,13], photocatalysis[14,15]. The optical characteristics of organic compounds fundamentally determine their applicability across diverse technological domains. For OLEDs, emission peak position and bandwidth directly impact color rendering and device efficiency[2]. For solar energy harvesting, compounds with absorption wavelengths matching the solar emission spectrum are especially valuable, making accurate prediction of this parameter crucial for identifying promising solar cell candidates without extensive experimental screening[16]. Similarly, in bioimaging applications, precise control over absorption wavelengths allows researchers to target specific "optical windows" where biological tissues exhibit minimal light attenuation, thereby enhancing imaging depth and resolution[10].

The optical properties of molecules are traditionally determined through experimental techniques such as UV/Vis spectroscopy[17]. While highly informative, these measurements can be time-consuming, resource-intensive, and limited in throughput, particularly when large chemical libraries or unstable compounds are involved. Moreover, the experimental identification of molecular structures with desired optical characteristics often relies on chemical intuition and iterative trial-and-error strategies, which are not always sufficient to explore the vastness of chemical space efficiently.

To address these challenges and to provide a more systematic understanding of molecular excited states, theoretical approaches have been developed to complement experimental methods. Among these, time-dependent density functional theory (TD-DFT) has emerged as a widely adopted tool for predicting the electronic structure of excited states and subsequently estimating absorption and emission wavelengths[18,19]. However, its accuracy is often limited by the approximate nature of exchange-correlation functionals, resulting in systematic deviations for charge-transfer transitions and double excitations[20,21]. In addition, the steep computational cost, which scales as $\mathcal{O}(M^3)$ with the number of basis wavefunctions $M$[22], makes TD-DFT impractical for large chromophores and high-throughput studies. Predicting more complex properties, such as photoluminescence quantum yield (PLQY) or emission lifetimes, requires modeling non-radiative decay and vibronic effects beyond the standard framework[23], while obtaining accurate absorption spectra demands consideration of vibrational broadening and multiple excited states, adding further computational complexity[24].

[1]AIRI, Moscow, Russia. [2]Moscow Institute of Physics and Technology, Moscow, Russia. [3]Lomonosov Moscow State University, Moscow, Russia. [4]Tomsk State University, Tomsk, Russia. [5]Kuban State University, Krasnodar, Russia. [6]ISP RAS Research Center for Trusted Artificial Intelligence, Moscow, Russia. [7]These authors contributed equally: Denis Potapov, Sergei Rogovoi. ✉e-mail: potapov@airi.net

In light of the limitations of TD-DFT, including accuracy constraints, high computational cost, and limited scalability, machine learning (ML) approaches have emerged as attractive alternatives[25–34]. These methods offer the potential to replicate or even surpass TD-DFT accuracy while enabling rapid screening across vast molecular libraries[25,30]. Studies employing algorithms such as random forests[35], support vector machines[36], and gradient boosting[37] with features such as circular fingerprints[38], Morgan fingerprints[39], and other SMILES-derived descriptors have shown that even simple representations can capture a substantial portion of the structural information needed to predict absorption properties[25–28]. At the same time, more advanced approaches based on deep residual convolutional neural networks[29] and graph neural networks (GNNs)[30–34] have been developed to leverage structured representations of molecules better to improve these results. For example, Greenman et al.[30] developed the UVVisML framework using a separate directed message passing network for dye and solvent. Sun et al.[32] integrated k-hop subgraphs, edge features, and global chemically intuitive features (e.g., measures of aromaticity, rotatable bonds) of chromophore into the GNN framework. Hung et al.[33] modified the SchNET model[40,41] by substituting traditional interatomic distance matrices with a bondstep representation derived from SMILES strings, effectively bypassing the collection of 3D conformations of chromophores. Fang et al.[42] utilized multi-modal LLMs and chain-of-thought finetuning. McNaughton et al.[34] addressed a related task of predicting the entire UV-vis absorption spectrum rather than just the peak positions. The study demonstrated that incorporating ab initio simulation data, such as 3D geometries of chromophores and quantum mechanically calculated spectra, alongside 2D molecular graph representations, enhances the prediction accuracy of experimentally obtained spectra. However, the performance boost is mainly due to the incorporation of quantum mechanically calculated spectra.

Recently, Neural Network Potentials (NNPs)[40,41,43–53] have emerged as an alternative to costly quantum mechanical simulations. NNPs have been applied, for example, to accelerate molecular dynamics simulations[54], conformational optimization[55,56], and solubility prediction[57]. To aid in training these models, large and diverse quantum chemical (QC) databases have been collected[58–67]. The abundance of high-quality data enables pretraining of NNPs that can be efficiently finetuned on more complicated datasets or different levels of theory[68,69], and to solve downstream molecular property prediction tasks[70–72].

Recent advances in quantum-chemically informed NNPs have inspired us to apply a similar approach to the prediction of optical properties. Among these, we focused on absorption peak wavelengths for several reasons. First, absorption data is significantly more abundant and diverse across datasets[25,73], providing a richer training signal. Second, the majority of existing machine learning models for molecular optoelectronic property prediction have concentrated on absorption[30,32,33,42], establishing it as a widely accepted benchmark. Finally, absorption is often the first experimentally accessible indicator of a molecule's electronic structure and is critical for a wide range of applications, from dye design to photovoltaics. Despite our primary focus on absorption peak wavelengths, this work also contains computational experiments on the prediction of other optoelectronic properties.

A major challenge in comparing machine learning models for chromophore optical properties is the lack of a standardized evaluation benchmark. Studies are based on disparate datasets: Lee et al.[31], Hung et al.[33], and Hung et al.[42] rely only on Deep4Chem[73]. Sun et al.[32] utilizes five datasets for testing. Deep4Chem, BODIPYs[74], JCIM_Abs[29], ChemFluor[25], and SMFluo1[75]. Greenman et al.[30] merge Deep4Chem, ChemFluor, CDEx, DyeAgg[76], and DSSCDB[77] into one large dataset. Evaluation protocols also differ. Lee et al.[31] and Sun et al.[32] use *k*-fold cross-validation. Greenman et al.[30], Hung et al.[33], and Fang et al.[42] perform a random train/validation/test split. Greenman et al.[30] demonstrates that naive random splitting can introduce data leakage: the same chromophore may appear in both training and test sets when solvent-solute pairs are treated as independent samples. They therefore suggest a scaffold split based on Bemis-Murcko scaffolds[78], which ensures that structurally similar molecules are grouped and restricted

to a single subset, thereby preventing overlap between training, validation, and test sets and enabling a more rigorous evaluation of model generalization.

To address the methodological and benchmarking challenges outlined above, our work makes the following key contributions:

- High-quality 3D dataset: We present nablaColors, a rigorously curated dataset of chromophore-solvent pairs with corrected SMILES and nablaColors-3D, a high-quality 3D conformations dataset optimized at multiple levels of theory (xTB, DFT in vacuum and implicit solvent). See section "Data preparation" for details.
- Benchmark for 3D GNN-based optical properties prediction: We introduce a dedicated benchmark specifically designed for evaluating 3D graph neural networks in molecular optical absorption prediction. This benchmark enables rigorous evaluation of both model architectures and conformational fidelity, as well as reproducible evaluation protocols for spatially informed methods.
- Systematic analysis of the effect of geometry optimization on prediction accuracy: We investigate how the accuracy of absorption prediction depends on the level of theory used for geometry optimization. This analysis enables us to quantify the trade-offs between computational cost and model performance.
- State-of-the-art absorption prediction with pretrained 3D GNNs: We evaluate four equivariant GNNs (PaiNN[79], DimeNet++[44], GemNet[45], eSCN[48]) and the molecular transformer UniMol+[80] and show that pretraining on large quantum-chemical datasets is critical for high performance. Our best model, combining pretrained UniMol+ with solvent embeddings, improves MAE by more than 30% compared to the strongest 2D GNN baseline evaluated under the same protocol.
- Multitarget optical property prediction with cross-validation: We extend the best-performing UniProp model to jointly predict absorption, emission, and PLQY in a multitarget setting using an expanded version of the dataset. We perform independent fivefold scaffold-based cross-validation for both single-target and multitarget setups, reporting strong performance across all properties, and establish the multitarget benchmark for solvent-aware 3D GNNs in optical property prediction.

## Results

In this section, we present benchmark results for 2D and 3D models trained on the curated nablaColors-3D dataset; full details of data preparation, model architectures, and training protocols are provided in the Methods section.

### Overall Model Performance

We evaluate all models on the proposed nablaColors-3D benchmark. Figure 1 shows the test set MAE (in nm) for all models, grouped by category.

Among the 2D baselines, Chemprop trained with explicit solvent information achieves the lowest error (23.60 nm), while other 2D models such as CGIB and Fluor-Predictor yield slightly higher MAE values. Despite not utilizing geometric information, these models demonstrate solid performance.

A consistent improvement is observed with 3D-based GNNs, which leverage spatial molecular structure. For each 3D model, we report the best result obtained through hyperparameter tuning on the validation dataset, using DFT-optimized conformers with implicit solvent and solvent embeddings introduced at the model regression head input. This configuration, providing both high-quality geometries and explicit solvent context, generally led to the lowest MAE and is used in all 3D models in this computational experiment.

Our modified UniProp model, which augments UniMol+ by incorporating a solvent embedding, delivers the best accuracy across all evaluated models, achieving, to the best of our knowledge, a state-of-the-art MAE of 15.97 nm for absorption wavelength prediction. To additionally contextualize these results against a first-principles baseline, we computed vertical absorption wavelength with TD-DFT (B3LYP/def2-

**Fig. 1 | Absorption MAE on the test split of our proposed nablaColors-3D benchmark for 2D and 3D GNNs.** For 2D GNNs we evaluate Chemprop variants[30], SOG[32], CGIB[31], and Fluor-Predictor[106]. For the TD-DFT baseline, the MAE is 62 nm. All models except the UniProp use geometries optimized with DFT and CPCM both during training and inference. UniProp model uses pairs of xTB-optimized, and DFT-optimized with CPCM model conformations during training, and xTB-optimized conformations during inference.

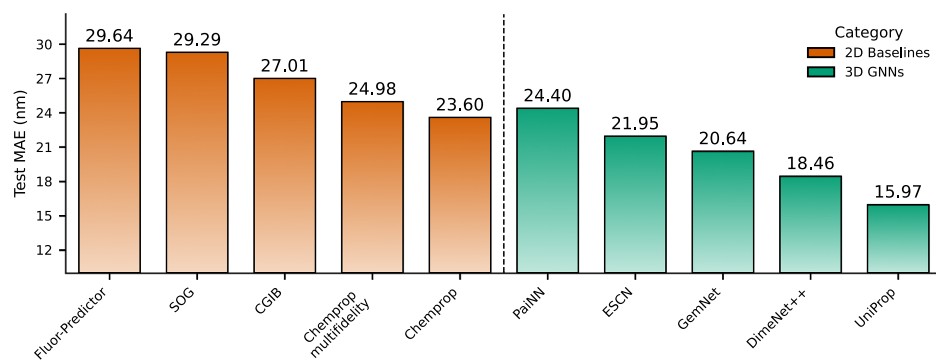

**Fig. 2 | Test MAE (nm) for five 3D GNN backbones across three conformer sources and two solvent regimes.** Colors encode conformer source: xTB, DFT in vacuum (DFT-vac), and DFT with implicit solvent via CPCM (DFT-imp). Hatching indicates models without a solvent embedding, whereas solid bars denote models with a Chemprop solvent embedding. All models except the UniProp use the specified type of conformations in both training and inference. UniProp only uses the specified conformation type in training. During inference, UniProp-xTB uses RDKit conformations, and UniProp-DFT-vac/DFT-imp use xTB conformations.

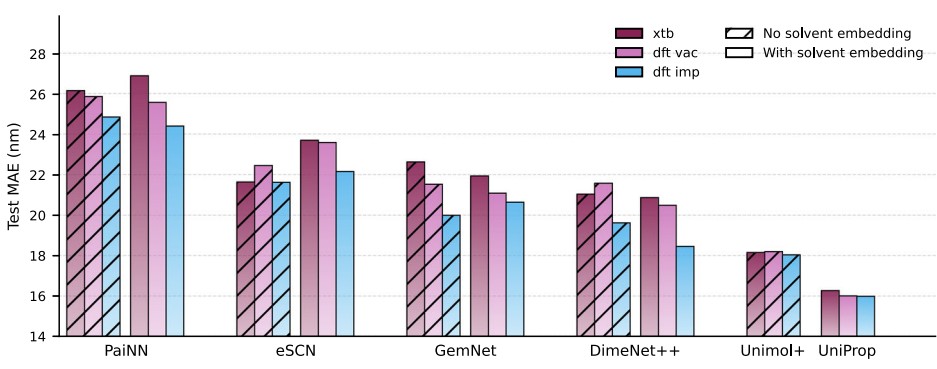

TZVPP[81,82] on the nablaColors-3D test set, obtaining an MAE of 62 nm. Full training curves for 3D GNNs can be found in Supplementary Fig. 1, and additional experiments on model finetuning strategies are in Supplementary Note 1.

The superiority of 3D GNNs over 2D models demonstrates that pre-training 3D GNNs on a large quantum-chemical dataset such as PubChemQC, together with the use of accurate conformers, provides a powerful inductive bias, enabling models to learn physically meaningful representations and substantially improving generalization for spectral property prediction.

### Effect of conformer fidelity

We study how the level of quantum theory used for geometry optimization affects optical property prediction by holding all training hyperparameters fixed and varying only the source of conformations. To separate architectural sensitivity from solvent information, we evaluate two regimes: models without any solvent embedding and the same backbones augmented with a Chemprop solvent embedding. Note that for UniMol+ and UniProp, the conformation choice influences only the training phase, not inference. Figure 2 summarizes test MAE across architectures and conformation sources.

Across architectures, higher-fidelity conformers are often associated with lower MAE, but the trend is not consistent for every model and task. In many cases, semi-empirical xTB geometries perform worse than DFT in vacuum geometries. Conformers optimized in implicit solvent further improve accuracy, especially when solvent embeddings are absent. We hypothesize that without explicit solvent embeddings, models extract the information about the solvent from chromophore conformations.

With solvent embeddings enabled, the gaps between different conformation types shrink substantially, suggesting that the embedding supplies much of the solvent-related signal the models require. For UniProp, final accuracy is largely insensitive to conformer fidelity, whereas for UniMol+ without solvent embeddings, higher-fidelity conformers generally yield better results. This yields a practical advantage: UniProp attains

### Table 1 | UniProp performance for different conformation-pair choices used during training

| Training pair | MAE | RMSE | $R^2$ |
|---|---|---|---|
| xTB → DFT implicit | 16.0 | 27.2 | 0.929 |
| RDKit → DFT implicit | 15.9 | 27.6 | 0.927 |
| RDKit → xTB | 16.3 | 27.8 | 0.926 |
| xTB → DFT vacuum | 16.0 | 27.8 | 0.926 |

We report row-wise MAE and RMSE (in nm) and the coefficient of determination $R^2$ on the test set.

strong performance without requiring costly high-level optimizations during training.

However, for UniProp trained on (xTB, DFT implicit) pairs, we still need to generate xTB-optimized conformations for inference, which introduces additional computational burden (see Supplementary Table 4). To mitigate this overhead, we conduct an additional experiment where we train UniProp on (RDKit, DFT implicit) conformation pairs directly. All training hyperparameters and the UniProp backbone are kept fixed. Only the source conformation is changed. The results in Table 1 show that (RDKit, DFT implicit) achieves essentially the same MAE as (xTB, DFT implicit), indicating that UniProp can maintain performance without the xTB optimization step during training.

### Multitarget learning and cross-validation

To further evaluate the generalizability of our UniProp model, we extended the prediction task from single-property regression to multitarget learning. Specifically, we trained the model to jointly predict the maximum absorption wavelength, the maximum emission wavelength, and the photoluminescence quantum yield (PLQY). This experiment was conducted on an extended dataset containing all three target properties. Since not all entries had complete annotations, we adopted a standard multitarget training approach using loss masking for missing values. Multitarget-

**Table 2 | Five-fold cross-validation MAE for UniProp and Chemprop**

| Model | Absorption only | Multitarget | | |
|---|---|---|---|---|
| | Abs MAE | Abs MAE | Ems MAE | PLQY MAE |
| Chemprop | 21.7 ± 2.3 | 22.0 ± 1.2 | 28.4 ± 1.4 | 0.17 ± 0.01 |
| UniProp | 14.8 ± 0.7 | 15.3 ± 1.0 | 19.7 ± 1.0 | 0.16 ± 0.01 |

Values are mean ± standard deviation across folds; absorption and emission in nm, PLQY unitless.

specific training details, can be found in Supplementary Methods and Supplementary Note 1. We additionally provide cross-validation results for the second best performing model, DimeNet++ with solvent embeddings, in Supplementary Table 3.

We evaluate single-target and multitarget setups using independent 5-fold scaffold-based cross-validation splits. In the single-target case, UniProp achieved an average MAE of 14.8 nm for absorption. In the multitarget setup, the model obtained average errors of 15.3 nm for absorption, 19.657 nm for emission, and 0.16 for log(PLQY). These results provide a baseline for multitarget optical property prediction with a 3D solvent-aware GNN and facilitate direct comparison with the 2D Chemprop baseline under the same protocol. The cross-validation results are provided in Table 2. During cross-validation, we identified one fold where both UniProp and Chemprop exhibited substantially higher errors. This degradation was linked to multiple measurements of the same solvatochromic probe (betaine dye 36, reported in ref. 83) across many unique, sparsely represented solvents. When these measurements fall into the test split, a pronounced distribution shift emerges, since most of the corresponding solvents appear only once in the dataset. For fair reporting, the cross-validation metrics in this work are computed with these entries removed from the affected test fold.

## Discussion

Our results establish that 3D pretrained GNNs provide a substantial improvement in absorption peak prediction compared to both 2D models and TD-DFT baselines. Our extensive benchmarking revealed that utilizing physically informed 3D representations substantially reduces prediction error, achieving more than 30% improvement in mean absolute error (MAE) relative to the strongest 2D baseline. This result is in line with previous research on molecular property prediction, in which 3D models outperform GNNs without explicit spatial features, even with fewer total parameters[80,84,85].

One of the factors behind this improvement is the high quality of molecular conformations used as input. Accurate atomic arrangements provided by high-level quantum chemical methods preserve subtle structural features such as bond angles, torsional preferences, and noncovalent interactions, all of which influence excited-state properties[86]. Specifically, the $r^2$SCAN-3c/def2-mTZVPP level of DFT employed here reliably reproduces interatomic distances and bond angles close to ground truth values[87]. By explicitly encoding this geometric information, 3D models generate physically meaningful molecular representations, substantially enhancing prediction accuracy. In contrast, 2D molecular graphs inherently lack this explicit geometric context, limiting their ability to capture geometry-dependent effects.

Compared to conventional quantum chemistry methods like TD-DFT, our pretrained 3D GNNs achieve comparable or better accuracy while reducing computational cost by several orders of magnitude. TD-DFT calculations scale poorly with molecular size[22] and require hours of CPU time per molecule (see Supplementary Table 4), whereas our models deliver predictions within milliseconds after a single conformer is generated. Among the evaluated 3D architectures, UniMol+ and our solvent-aware variant, UniProp, achieved the most accurate predictions, substantially outperforming PaiNN, DimeNet++, GemNet, and eSCN. While the cost of generating 3D conformations for UniMol+ is higher compared to 2D models operating on SMILES alone, its superior accuracy far outweighs this additional cost, making it a more reliable choice for applications requiring high predictive fidelity. The key advantage of UniMol+ lies in its dual

capability to both refine molecular geometries and learn optical properties simultaneously, which most other GNN-based models do not explicitly address. First, the SE(3)-invariant transformer backbone with global attention allows UniMol+ to capture long-range interatomic correlations that are critical for electronic transitions but often missed by GNNs with a limited receptive field. Transformers' global receptive field, coupled with 3D positional encodings (spatial and graph-based), ensures a comprehensive structural representation. Second, the coordinate-denoising pretraining paradigm, where raw conformations from the low-cost computational method are iteratively updated toward low-energy DFT-optimized geometries, provides UniMol+ with an implicit understanding of the energy landscape. The model learns a quasi-optimization trajectory (akin to a few-step surrogate for DFT relaxation), enabling accurate property prediction even from approximate input geometries. This is particularly beneficial when high-level conformations (e.g., from DFT) are unavailable at inference time. The UniMol+ paper[80] shows that this conformation-refinement capability contributes directly to its superior scores on PCQM4MV2[88] and OC20[89].

This design has important practical consequences. Since UniMol+ internalizes a correction mechanism for approximate geometries during pretraining, high-fidelity DFT conformations are only required in the training phase. This results in a substantial reduction of computational cost during inference compared to TD-DFT and 3D GNNs tested in this work, as UniMol+ can operate effectively on inexpensive xTB conformers. Such efficiency makes it feasible to perform large-scale virtual screening on datasets containing tens or hundreds of thousands of molecules, which would otherwise be impractical with DFT-level geometries.

Another substantial aspect influencing model performance was careful data curation. By identifying and correcting problematic entries, such as inconsistent protonation states, invalid SMILES strings, and incorrect metal coordination patterns, we substantially reduced label noise. Data denoising is widely recognized as a crucial technique for improving predictive accuracy and preventing overfitting, as demonstrated in prior studies on molecular property prediction[90–95]. Based on empirical observations during dataset construction and early model development, we found that rigorous data cleaning was essential for stable training and competitive performance, consistent with prior reports emphasizing the importance of high-quality molecular datasets.

Incorporation of explicit solvent embeddings via a lightweight 2D encoder provided additional predictive gains. This solvent-aware module explicitly models molecular interactions with solvents, capturing shifts in molecular geometry and electronic structure induced by the solvent environment. Notably, even with solvent embeddings provided to all models, only those using solvent-optimized geometries generally outperformed others. This underscores the importance of explicitly modeling solvent-induced structural adjustments beyond mere solvent descriptors, suggesting further exploration of solvent-aware geometry optimization for future research.

Beyond absorption prediction, the proposed solvent-embedding architecture offers a general mechanism for learning solvent-specific effects in various property prediction tasks. For the 3D GNNs tested here, the same approach could be extended to predict energies and forces in the context of NNPs, reaction barriers, or solvation-dependent properties. UniMol+, initially trained to refine geometries only in vacuum, can now be adapted to predict solvent-adjusted molecular structures.

The multitarget learning experiments provide compelling evidence for the robustness and versatility of our 3D GNN approach. While we observed a modest increase in absorption prediction error when transitioning from single-target to multitarget learning, this minimal degradation demonstrates that our model maintains its predictive accuracy even when learning multiple optical properties simultaneously. The cross-validation results further validate the stability of our approach, showing consistent performance across different data splits despite the inherent challenges of multitarget learning, including varying data availability across properties and potential interference between learning objectives. Notably, the ability to

jointly predict absorption wavelength, emission wavelength, and photoluminescence quantum yield from a single model represents a significant practical advantage for molecular design applications. This multitarget capability transforms our approach from a specialized absorption predictor into a comprehensive optical property prediction platform, enabling researchers to efficiently screen molecular candidates across multiple optical characteristics without requiring separate models for each property. The successful extension to emission wavelength (19.894 nm MAE) and PLQY (0.156 MAE) prediction demonstrates that the 3D geometric representations learned by UniProp capture fundamental molecular features relevant to diverse photophysical processes, not merely absorption-specific electronic transitions.

Despite these promising results, several limitations and opportunities for further development exist. A detailed error analysis (Supplementary Note 2) indicates that residual errors are more strongly associated with out-of-distribution chromophores (as measured by training-set similarity) than with simple size/flexibility descriptors; moreover, a targeted conformer-ensemble test using CREST[96] did not yield systematic improvements for the highest-error molecules. The solvent embedding MPNN block used in this study, while effective, was intentionally kept relatively small due to dataset constraints, including limited solvent diversity and dataset size. Future studies might benefit from larger and more diverse datasets, enabling the exploration of more expressive solvent encoding architectures. Furthermore, while UniMol+ provides robustness and superior predictive accuracy, its computational complexity and memory demands may restrict scalability to extremely large molecules or extensive conformational ensembles. Efforts to optimize these architectures for broader applicability represent a compelling direction for future research.

In conclusion, our findings establish the pronounced advantage of employing 3D GNNs, particularly those incorporating global attention mechanisms and geometry-aware pretraining strategies, for predicting molecular optical properties. By meticulously curating high-quality datasets, explicitly modeling solvent effects, and leveraging multi-level geometric fidelity during model training, we provide a robust and efficient methodology that substantially surpasses traditional methods and allows for accurate virtual screening of optically active molecules.

## Methods
### Data preparation
To obtain the proposed nablaColors and nablaColors-3D datasets, we extended the dataset integration strategy of Greenman et al.[30] by incorporating not only maximum absorption wavelengths, but also maximum fluorescence wavelengths and photoluminescence quantum yields (PLQYs). To construct a unified dataset, we combined data from Deep4Chem[73], ChemFluor[25], DyeAgg[76], and DSSCDB[77]. Following Greenman et al.[30], we excluded the CDEx dataset[97] due to concerns about data quality.

During the construction of the combined dataset, we encountered several data quality issues. These included invalid SMILES strings, redundant entries with multiple protonation states assigned to what was nominally the same molecule, and broken $\pi$-conjugated systems—particularly in squaraine and cyanine dyes—where incorrect SMILES disrupted the conjugated backbone. A notable source of inconsistency came from entries in ref. 25, where molecules were sourced from online databases in solvated, often charged forms (including counterions). These structures were included in the dataset with varying protonation states but were assigned identical optical property values, despite representing chemically distinct species. Further complications arose when handling metal-organic compounds. Their correct protonation was challenging to determine due to the limitations of the SMILES format, which cannot fully capture coordination environments. Conformer generation for metal-porphyrins also posed a challenge, as SMILES cannot represent donor-acceptor and coordination bonds. In some cases, geometry optimization led to bond rearrangements, producing molecular graphs that were no longer isomorphic to those derived from the original SMILES.

To correct such inconsistencies, we manually verified problematic entries by consulting the original experimental studies cited in the source datasets. This verification allowed us to reconstruct accurate molecular structures where possible and ensure consistency between structural representations and associated optical properties. Through careful inspection of the dataset, we removed 1825 problematic entries and replaced 60 with corrected structures. We refer to the resulting dataset as nablaColors.

Many state-of-the-art machine learning models in chemistry require accurate 3D molecular conformations in addition to the molecular graph structure for reliable property prediction[80,84,98]. In this work, we established the following computational pipeline to obtain optimized molecular conformations from the SMILES strings available in our filtered nablaColors dataset:•  SMILES → initial 3D. RDKit's ETKDG2 algorithm[99] generates a 3D Cartesian geometry, which is then optimized with the MMFF94 force field[100].

- xTB geometry optimization. The MMFF-optimized geometry is further optimized at the GFN2-xTB level of theory in vacuum with the xTB package[101]. The xTB optimization takes 0.37 CPU-hours per molecule on average.
- DFT optimization. The xTB-optimized structure is refined with density functional theory (DFT) at the r²SCAN-3c/def2-mTZVPP[102] level of theory implemented in ORCA[103]. We perform the DFT optimizations both in vacuum (15 CPU-hours per molecule on average) and with the Conductor-like Polarizable Continuum Model (CPCM)[104] (20 CPU-hours per molecule on average).

The r²SCAN-3c/def2-mTZVPP level of theory was chosen for DFT geometry optimizations following established best-practice recommendations for molecular structure determination. This composite meta-GGA approach has been shown to provide accurate and robust equilibrium geometries, reproducing bond lengths and angles while remaining computationally efficient and well suited for large-scale studies. As summarized by Bursch et al.[105], r²SCAN-3c represents an optimal cost-accuracy compromise for routine geometry optimization tasks.

CPU times required for each optimization stage are reported in Supplementary Table 4.

Throughout conformation preparation, we carefully verified that the molecular graph topology of each optimized structure remained consistent with the original SMILES representations. Additionally, during the curation process, we identified several duplicate entries and erroneous data points, which were either corrected or removed based on examination of the original literature sources. This process resulted in 13731 individual molecules and 26369 chromophore-solvent pairs. For each chromophore-solvent pair, the dataset contains 4 conformations calculated at different levels of theory. We refer to this curated set as nablaColors-3D. The data processing pipeline is visualized in Fig. 3.

### 3D-GNNs for peak absorption wavelength prediction
State-of-the-art models for optical-property prediction are still dominated by 2D GNNs such as Chemprop[30], SOG[32], CGIB[31] and Fluor-Predictor[106]. However, the 3D molecular conformation modulates the electronic structure[86,107] and thus the absorption spectrum. Incorporating three-dimensional information into the GNN has therefore been shown to yield systematic performance gains[84,88,98].

To explore this advantage, in this work, we benchmark five GNNs: PaiNN[79], DimeNet++[44], eSCN[48], GemNet-OC[45], and UniMol+[80]. PaiNN, DimeNet++, eSCN, and GemNet-OC were chosen because they deliver the highest accuracy on the recent ∇²DFT benchmark[63], whereas UniMol+ is one of the top performers on the PCQM4Mv2 leaderboard[88]. PCQM4Mv2 is a large-scale quantum chemistry dataset that provides DFT-optimized geometries and HOMO-LUMO gaps, which are strongly correlated with absorption maxima[108]. We therefore expect pretraining on this dataset to yield transferable representations for the optical property prediction task. Accordingly, we pretrained PaiNN, DimeNet++, eSCN, and GemNet-OC on PCQM4Mv2 and initialized UniMol+ from its official checkpoint. All

**Fig. 3 | Data processing pipeline.** First, the UV/Vis dataset of absorption spectra is curated to obtain the nablaColors dataset. Then we generate chromophore conformations for each unique chromophore in nablaColors. After that, the chromophore conformations are sequentially optimized with MMFF first, then with xTB (I), and finally with DFT in both vacuum (II) and implicit solvent (III). We refer to the resulting collection of 3D conformations as nablaColors-3D.

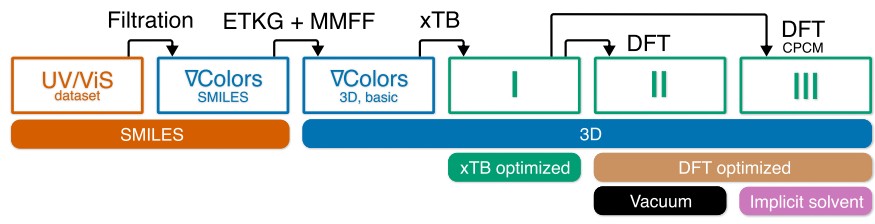

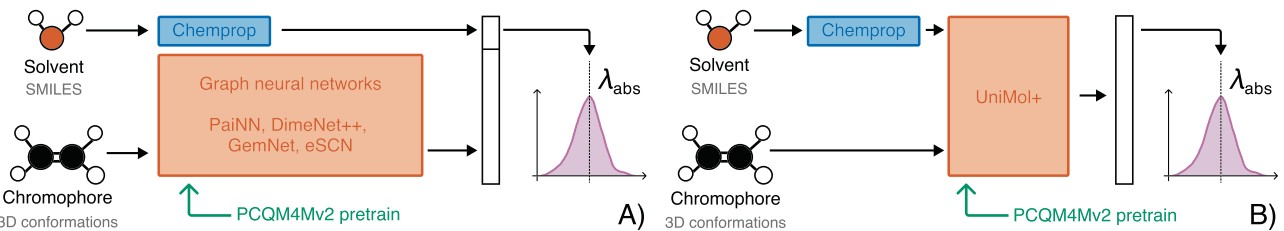

**Fig. 4 | Architecture of 3D GNNs for optical properties prediction.** The solvent is represented as SMILES and processed with a Chemprop model, and A) solvent embeddings are concatenated with final chromophore embeddings obtained for PaiNN, DimeNet++, GemNet, and eSCN, B) solvent embedding added to the virtual node for UniMol+ prior to the self-attention blocks. The predicted target, $\lambda_{abs}$, denotes the wavelength corresponding to the maximum of the absorption spectrum (absorption maximum).

five models were then finetuned on nablaColors-3D to predict experimental peak absorption wavelengths. Full details of the pretraining and finetuning procedures, as well as the evaluation protocols, are provided in Supplementary Methods.

The role of the solvent is substantial but secondary relative to the chromophore geometry. Solvent contribution explains only a minor fraction of the overall variance in the dataset. Therefore, we deliberately avoid heavy 3D modeling of the solvent and instead test two complementary strategies: (1) using geometries optimized with an implicit solvent model, and (2) enriching the chromophore representation with a solvent embedding.

For the second approach, we combine a 3D GNN backbone for the chromophore with a lightweight 2D Chemprop encoder[109] for the solvent. This choice is motivated by the relatively small set of unique solvents in the dataset, which does not justify the use of a complex model with a large parameter count. A message-passing architecture like Chemprop provides a sufficiently expressive yet parameter-efficient approximation of solvent effects.

The Chemprop encoder was pre-trained on SMILES strings from the training subset of nablaColors to predict optical properties, using hyperparameters optimized for this task. In this pretraining stage, each sample corresponds to a chromophore-solvent pair: the chromophore SMILES and the solvent SMILES are processed by two Chemprop MPNNs with identical architectures and hyperparameters, but independent weights. The resulting chromophore and solvent latent embeddings are concatenated and passed to a regression head to predict the, peak absorption wavelength. After pretraining, we reuse the solvent Chemprop MPNN as a fixed source of solvent embeddings in our 3D-GNN experiments. During 3D model training and evaluation, only the solvent SMILES is passed through this pretrained Chemprop encoder to obtain a solvent embedding, which is then combined with the chromophore representation produced by the 3D GNN backbone (Fig. 4A). This design allows the 3D GNN to focus on capturing geometric and electronic effects of the chromophore, while the Chemprop encoder provides a compact and data-efficient representation of solvent effects. Implementation details, including the Chemprop version used in this study, are provided in Supplementary Methods.

Training is performed in two phases: in the first phase, both pretrained encoders are frozen while the regression head is trained to convergence; in the second phase, the entire network, including both Chemprop and the 3D GNN backbone, is unfrozen for a joint finetune. Full training schedules, architectural variants, optimization hyperparameters, and experimental setup details are provided in Supplementary Information Table S1.

## UniMol+ for peak absorption wavelength prediction

UniMol+[80] incorporates a geometry-refinement objective during pretraining. Each training instance comprises a low-cost input conformation (e.g. xTB-optimized) and a high-accuracy reference geometry optimized at the DFT level of theory. The network is trained jointly to predict (i) reference coordinates from the input coarse structure (see Lu et al.[80] for details) and (ii) the target property (i.e., peak absorption wavelength). By learning to map cheap input conformers to DFT-quality geometries during training, this approach retains the accuracy associated with DFT-optimized structures while using only low-cost at inference. This feature enables us to avoid the computational bottleneck caused by conformation generation that constrains 3D GNNs, which achieve their best accuracy only with DFT-optimized inputs.

We propose UniProp, a solvent-aware variant of UniMol+ in which the 2D Chemprop encoder is combined with the UniMol+ transformer backbone to incorporate solvent effects. After projecting the Chemprop-derived solvent embedding onto the atom feature dimension, we sum it with the virtual node embedding, a specialized token analogous to the [CLS] token in BERT[110] (Fig. 4B), used as a global molecular representation for property prediction. This allows solvent effects to influence the intermediate molecular representations within the model, rather than being introduced only at the final prediction stage. Such integration aligns with chemical intuition, as solvent interactions impact both the molecule structure and spectral properties of the chromophore, leading to a notable improvement in prediction accuracy. This UniProp architecture delivers the best performance according to our benchmark, outperforming the best 2D model by more than 30% in MAE (see Fig. 1).

To further evaluate the robustness of our results, we performed 5-fold scaffold-based cross-validation for UniProp on the single-target absorption task, using a dataset containing only absorption annotations. This cross-validation was run independently from that used for the multitarget setting, ensuring that both evaluations reflect performance under comparable but non-overlapping data splits.

## Benchmarking setup

Despite significant progress in machine learning for molecular property prediction, the field has lacked a unified and high-quality benchmark for

optical and photoluminescence properties. Existing datasets often suffer from corrupted SMILES strings or lack 3D geometries, making it difficult to assess model performance under realistic and controlled conditions.

To address this gap, we introduce a new benchmark based on the nablaColors-3D dataset, which combines experimental measurements of absorption maxima, emission maxima, and photoluminescence quantum yields with optimized molecular geometries across multiple levels of quantum chemical theory, including solvent-aware methods. This benchmark is designed to facilitate rigorous and fair evaluation of both model architectures and conformer fidelity for spectral property prediction.

We follow the scaffold split strategy proposed in Greenman et al.[30], using Bemis-Murcko scaffold partitioning to ensure that structurally similar molecules do not appear in both training and evaluation sets. The dataset is divided into training, validation, and test splits in an 80%/10%/10% ratio. The validation set is used to select hyperparameters and choose the best-performing model checkpoint. Performance is evaluated using Mean Absolute Error (MAE, in nanometers) on the held-out test set.

We benchmark several state-of-the-art 2D graph neural networks (GNNs) for absorption peak prediction, including Chemprop, Chemprop-multifidelity[30], SOG[32], CGIB[31], and Fluor-Predictor[106]. All 2D models (except for Chemprop, see Section "3D-GNNs for peak absorption wavelength prediction" for details) were trained from scratch using hyperparameters from their official implementations. As a reference for physically grounded prediction, we include TD-DFT results for the test set computed at the B3LYP/def2-TZVPP[81,82] level of theory with implicit solvent modeling.

Finally, to investigate how the quality of molecular geometry affects prediction accuracy, we trained a separate 3D GNN for each of the three conformer types in our dataset. Each model was trained using identical splits and hyperparameters, allowing us to isolate the effect of conformer fidelity on model performance.

## Multitarget property prediction

To further evaluate the robustness of our best-performing 3D architecture from the single-target benchmark, we tested its performance in a multitarget setting where the model was trained to jointly predict maximum absorption wavelength, maximum emission wavelength, and photoluminescence quantum yield (PLQY). We used the solvent-aware UniProp variant of UniMol+ with the same architectural configuration and hyperparameters as in the single-target absorption experiments (see Section "To further evaluate the robustness of our best-performing"); multitarget-specific settings and training schedule are given in Supplementary Methods. Missing labels in the extended dataset were handled via loss masking, ensuring that gradients were computed only for available annotations.

For both the multitarget and single-target cases, we independently applied a 5-fold scaffold-based cross-validation protocol. Performance was evaluated on the held-out folds using MAE for each property reported in the Supplementary Table 3. For reference, we also trained Chemprop[30] on the same multitarget dataset using the same cross-validation setup. Chemprop was chosen as the 2D model with the strongest performance from previous optical property studies. This comparison serves to contextualize UniProp's performance in the multitarget regime relative to a competitive 2D baseline.

## Data availability

The dataset and model checkpoints supporting the findings of this study are openly available at https://zenodo.org/records/18061300[111].

## Code availability

The associated code, The code supporting the findings of this study is openly available on GitHub at https://github.com/AI4DD/nablaColors[112].

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

## Acknowledgements

This work was supported by a grant, provided by the Ministry of Economic Development of the Russian Federation in accordance with the subsidy agreement (agreement identifier 000000C313925P4G0002) and the agreement with the Ivannikov Institute for System Programming of the Russian Academy of Sciences dated June 20, 2025 No. 139-15-2025-011.

## Author contributions

D.P.: Data curation, Validation, Conceptualization, Investigation, Software, Formal analysis, Visualization, Writing–original draft. S.R.: Data curation, Formal analysis, Investigation, Software, Writing–original draft. K.K.: Data curation, Investigation, Writing–review & editing. K.U.: Visualization, Writing–review & editing. A.K.: Writing–review & editing. A.B.: Data curation. A.K.: Supervision, Resources, Writing–review & editing. A.T.: Supervision, Conceptualization, Formal analysis, Software, Visualization, Writing–original draft.

## Competing interests

The authors declare no competing interests.
