## [Transparent Peer Review file · Communications Chemistry]

A Conformational Benchmark for Optical Property Prediction with Solvent-aware Graph Neural Networks

Corresponding Author: Mr Denis Potapov

Version 0:

Reviewer comments:

Reviewer #1

(Remarks to the Author)

#####

Summary:

The authors address the well-known challenge of molecular optical property prediction by creating a new benchmark dataset constructed by curating existing datasets and cleaning them rigorously, generating 3D conformations of these chromophores at several levels of theory, and then benchmarking a series of 2D and 3D GNNs on this dataset. They incorporate solvent effects and demonstrate that their best performing 3D model shows substantial performance gains over the existing state of the art.

#####

Strengths:

Overall, I think this is excellent work that creates a useful benchmark dataset for the community, both by cleaning and combining existing datasets more carefully than previous work has done, and by generating and publicly releasing 3D structures for all of these chromophores. The authors thoroughly evaluate the state-of-the-art 2D methods and compare them with some of the most popular 3D architectures (with their own addition of a 2D chemprop model to embed solvent information). They perform experiments that allow them to disaggregate the effects of the conformer fidelity from the effects of the model architecture choice. The authors use appropriately rigorous splitting of the dataset. They perform cross validation when comparing the best 2D and 3D models to one another to be more rigorous, and they expand their evaluation to include the multitask setting of predicting emission and PLQY in addition to absorption. The dataset has been uploaded to Zenodo, and the code for the best performing model has been made public on GitHub.

#####

Major questions / suggestions:

- Can the authors expand on what they mean by "The Chemprop encoder was pre-trained on SMILES strings from the training subset of ∇ Colors to predict optical properties, using hyperparameters optimized for this task." What was the target used for Chemprop during this "pretraining" task? When you say "SMILES strings from the training subset of ∇ Colors", do you mean chromophore SMILES or solvent SMILES? It seems like it should be pre-trained on solvent SMILES since that's what it will eventually be used for, but it's not clear to me what the task would be for this.

- I suggest the authors add DimeNet++ (the 2nd-best-performing 3D model) to the evaluation in Table 1, because based on the performance in Figs 4 and 5, it seems possible that the difference in performance between UniProp and DimeNet++ might overlap if subjected to cross validation.

- The authors should also report RMSE and/or R² metrics for at least some of the tasks, as the use of more than one metric

provides a more holistic evaluation of a model's performance and is a best practice for ML in chemistry (G. Vishwakarma, A. Sonpal, J. Hachmann, Trends Chem. 3, 146–156 (2021)).

- Could the authors provide some analysis of whether there are any trends of which molecules are benefiting most from including 3D molecular information, or which are still predicted poorly despite having 3D information? I suspect this may depend on the size and/or flexibility of the chromophore, or the family to which it belongs. Are there any classes of chromophores in the benchmark dataset where your analysis indicates that it may be necessary to generate an ensemble of conformers (4D information) in order to accurately predict the optical properties? Or where you suspect that a local minimum found by conformer generation leads to an incorrect prediction?

- In the Discussion, you say "While the cost of generating 3D conformations for UniMol+ is higher compared to 2D models operating on SMILES alone, its superior accuracy far outweighs this additional cost, making it a more reliable choice for applications requiring high predictive fidelity." What is the approximate difference in inference speed (e.g. average per 1000 molecules) for the various 2D and 3D methods? They're all obviously much faster than TD-DFT, and it's great how you show that UniProp can perform well with only the xTB geometry as input at inference time, but it seems like the extra cost of running xTB (avg 27.54 s/molecule) plus the use of equivariant 3D architectures could still make your best method noticeably slower than the 2D methods. You may be correct that the additional cost could be worthwhile, but I think it's difficult to make a cost-benefit comparison like this without actually quantifying the costs, and quantifying this could help future researchers evaluate this cost-benefit tradeoff for themselves. Additionally, since the 3D methods other than UniProp require the DFT geometries at inference time, providing some timings would further strengthen your case for UniProp as the best choice (though it seems that for most of the other 3D models, using a DFT geometry instead of xTB only reduces MAE by ~1-3nm anyway, and in some cases actually makes it worse when there is no solvent embedding).

- Several places throughout the manuscript, the authors use the word "significantly" but do not perform any statistical significance tests. For example, in the Discussion, "Among the evaluated 3D architectures, UniMol+ and our solvent-aware variant, UniProp, achieved the most accurate predictions, significantly outperforming PaiNN, DimeNet++, GemNet, and eSCN." With no cross-validation comparison for the 3D models and no statistical test, I think it would be more appropriate to use a different word (e.g. "substantial"). The authors also use the word "significantly" several times when comparing 2D to 3D models; although they do cross-validation and report error bars when comparing Chemprop and UniProp, they do not perform any significance test. If the authors would like to make a claim of statistical significance, I suggest they consult this recent paper for best practices: <https://pubs.acs.org/doi/full/10.1021/acs.jcim.5c01609>.

- It appears from the GitHub that the authors have provided the scripts to reproduce the training and inference of their top performing UniProp method. In addition to this, I encourage the authors to also publish (on Zenodo) the trained version of their model so people can easily use it without having to retrain it. It would be even better if it's possible to provide a script to automatically generate the RDKit 3D  xTB conformers and store these in the correct format, so the only thing a user needs to provide to run inference is a list of chromophore SMILES and corresponding solvent SMILES.

#####

Minor questions / suggestions:

- The statement "we filtered 1885 chromophores, either by removing problematic entries or replacing them with corrected structures" is a bit ambiguous, and when I first read it I thought it meant that there were only 1885 chromophores left after filtering out the problematic entries. This is particularly confusing because you don't mention the "13731 individual molecules and 26369 chromophore-solvent pairs" until later in the next paragraph when you're describing the 3D dataset. After inspecting the Zenodo files, I think it could be more clear to say here "we removed 1825 problematic entries and replaced 60 with corrected structures" or something like that, and to state the total number remaining here as well.

- I think a brief justification of the choice of DFT functional / basis set for geometry optimizations (r2SCAN-3c/def2-mTZVPP) is warranted in the Methods. The authors mention some justification in the Discussion section ("Specifically, the r2SCAN-3c/def2-mTZVPP level of DFT employed here reliably reproduces interatomic distances and bond angles close to ground truth values").

- It would be helpful for you to specify which major version of Chemprop you have used for the study, because the default featurizer and some other things changed between v1.x and v2.x. Additionally, the models trained using v1.x are not compatible with v2.x without first running a conversion script, so I suggest the authors also add this version specification in their GitHub repository where the trained Chemprop model is stored.

- I suggest the authors emphasize the performance of the TD-DFT B3LYP/def2-TZVPP baseline (62 nm) more, maybe by adding it to the caption of Fig 4 or Fig 5, since this is a key point that all of these methods perform much better than TD-DFT at a fraction of the cost. Normally I might suggest adding a horizontal line for the TD-DFT value for comparison on the bar plots in Fig 4 and 5, but the value of 62nm is so much larger than the ML models that it would distort the rest of the plot, so I think a comparison in the caption could be helpful for someone just skimming the paper.

- I think the captions of Fig. 4 and 5 could mention what level of geometry was used at training and at inference for each of these 3D methods. If I understand correctly, in Fig 4., all of these used DFT with implicit solvent during training, all but UniProp used DFT with implicit solvent during inference, and UniProp used xTB at inference. And in Fig. 5, the different

colors of the bars indicate which geometry level is used for training, and the same level is used at inference except in the case of UniMol+ where xTB is used at inference.

- Do I also understand correctly that in Fig. 5, for the right most set of 6 bars, the 3 hatched bars are UniMol+, and the 3 unhatched bars are UniProp, even though you label them collectively as Unimol+ (since UniProp is UniMol+ with the Chemprop solvent embedding)?

- I think the use of 5 significant figures in Table 1 and the following paragraphs is probably overkill (1 or 2 digits after the decimal place should be sufficient for values in nanometers).

- When you describe why you dropped some entries from the dataset during cross validation, you say "During cross-validation, we found that UniProp consistently underperformed on one specific split." I suggest mentioning that Chemprop also performed much worse on these same splits, which your results in Table S3 demonstrate. Without saying that, it may sound like the UniProp model is uniquely bad at capturing the solvatochromism of betaine dye36 in many unique solvents.

- The authors mention in the Discussion that "In our preliminary experiments, rigorous cleaning not only reduced the MAE but also enhanced training stability, underscoring the importance of high-quality datasets when training expressive 3D GNNs." Are you able to provide any data in the SI to support these claims?

- Why did the authors choose to model PLQY in addition to absorption and emission wavelength, but not FWHM bandwidth or other properties that are present in some of the source datasets?

- I suggest that the authors specify how the error bars in Table 1 were calculated (standard error?).

- In the abstract and Introduction, the authors mention an improvement of "more than 30%" over the previous best 2D model, but section 2.3 and the Discussion say "nearly 25%".

- In Table S2, should the "Pretrained" be changed to "Staged" to match the nomenclature described in section S1.3?

- In Table S3, should "Mean Val Loss" be "Mean Test Loss"?

#####

Typos / formatting:

- Inconsistent capitalization of "Chemprop" vs. "ChemProp" throughout, including in the text, tables, captions, and figures. I suggest the authors use the canonical capitalization "Chemprop" (change all "ChemProp" instances to "Chemprop").

#####

Reviewer #2

(Remarks to the Author)

The work in its current form is a benchmark report and a demonstration of model scaling. The manuscript is written mainly for the prediction of optical properties by using 3D GNN models in a solvent-aware manner, combined with the curated dataset and optimized molecular geometries in different quantum mechanical levels as input. Considering that too many model works have been done to improve the prediction accuracy, the manuscript fails to provide more physical insights regarding the absorption and emission properties forecasting, except for the MAE reduction.

I therefore cannot recommend the publication of this manuscript in its current form.

Version 1:

Reviewer comments:

Reviewer #1

(Remarks to the Author)

I have reviewed the point by point response letter and am satisfied with the authors' response; I now recommend that this manuscript be accepted for publication.

Reviewers' comments:

Reviewer #1 (Remarks to the Author):

#####

Summary:

The authors address the well-known challenge of molecular optical property prediction by creating a new benchmark dataset constructed by curating existing datasets and cleaning them rigorously, generating 3D conformations of these chromophores at several levels of theory, and then benchmarking a series of 2D and 3D GNNs on this dataset. They incorporate solvent effects and demonstrate that their best performing 3D model shows substantial performance gains over the existing state of the art.

#####

Strengths:

Overall, I think this is excellent work that creates a useful benchmark dataset for the community, both by cleaning and combining existing datasets more carefully than previous work has done, and by generating and publicly releasing 3D structures for all of these chromophores. The authors thoroughly evaluate the state-of-the-art 2D methods and compare them with some of the most popular 3D architectures (with their own addition of a 2D chemprop model to embed solvent information). They perform experiments that allow them to disaggregate the effects of the conformer fidelity from the effects of the model architecture choice. The authors use appropriately rigorous splitting of the dataset. They perform cross validation when comparing the best 2D and 3D models to one another to be more rigorous, and they expand their evaluation to include the multitask setting of predicting emission and PLQY in addition to absorption. The dataset has been uploaded to Zenodo, and the code for the best performing model has been made public on GitHub.

#####

Major questions / suggestions:

- Can the authors expand on what they mean by "The Chemprop encoder was pre-trained on SMILES strings from the training subset of ∇ Colors to predict optical properties, using hyperparameters optimized for this task." What was the target used for Chemprop during this "pretraining" task? When you say "SMILES strings from the training subset of ∇ Colors", do you mean chromophore SMILES or solvent SMILES? It seems like it should be pre-trained on solvent SMILES since that's what it will eventually be used for, but it's not clear to me what the task would be for this.

The Chemprop encoder was pre-trained on SMILES strings from the training subset of \dataset{} to predict optical properties, using hyperparameters optimized for this task.

Ans: In our implementation, the term “Chemprop pretraining” refers to training a standard 2D Chemprop model on the training split of the dataset using the same supervised objective as in the downstream task.

*Each training sample corresponds to a chromophore–solvent pair. The chromophore SMILES and the solvent SMILES are encoded using **two Chemprop MPNN encoders with identical architectures and hyperparameters but independent weights**. The resulting chromophore and solvent embeddings are concatenated and passed to a regression head to predict the target optical property (peak absorption wavelength).*

*After this supervised training stage, we reuse the **solvent Chemprop encoder** as a source of solvent embeddings in the 3D-GNN experiments. During 3D model training and evaluation, only the solvent SMILES is passed through this pretrained encoder to obtain a latent solvent representation, which is then combined with the chromophore representation produced by the 3D backbone (via concatenation at the prediction head for PaiNN, DimeNet++, GemNet, and eSCN, or via injection into the virtual token for UniProp).*

We have added a clarification in Section 2.2 to explicitly describe the pretraining target, the use of both chromophore and solvent SMILES, and the use of separate Chemprop encoders with independent weights.

- I suggest the authors add DimeNet++ (the 2nd-best-performing 3D model) to the evaluation in Table 1, because based on the performance in Figs 4 and 5, it seems possible that the difference in performance between UniProp and DimeNet++ might overlap if subjected to cross validation.

Ans: We have cross validated the DimeNet++ with solvent embeddings and included the results in Supplementary Section 2.4. DimeNet++ performs worse than UniProp on all folds.

- The authors should also report RMSE and/or R^2 metrics for at least some of the tasks, as the use of more than one metric provides a more holistic evaluation of a model's performance and is a best practice for ML in chemistry (G. Vishwakarma, A. Sonpal, J. Hachmann, Trends Chem. 3, 146–156 (2021)).

Ans: We now report the suggested metrics for different variants of the UniProp model in Table 1.

- Could the authors provide some analysis of whether there are any trends of which molecules are benefiting most from including 3D molecular information, or which are still predicted poorly despite having 3D information? I suspect this may depend on the size and/or flexibility of the chromophore, or the family to which it belongs. Are there any classes of chromophores in the benchmark dataset where your analysis indicates that it may be necessary to generate an ensemble of conformers (4D information) in order to accurately predict the optical properties? Or where you suspect that a local minimum found by conformer generation leads to an incorrect prediction?

Ans: We thank the reviewer for this insightful question regarding which molecules benefit most from including 3D information and which remain challenging despite having 3D geometries.

*To address this point, we performed a dedicated post-hoc error analysis on the held-out test set and added a new **Supplementary section S3**. In this analysis, we examined correlations between per-molecule UniProp prediction error and improvement over Chemprop with several molecular descriptors and dataset-coverage proxies, including molecular weight, number of rings, number of rotatable bonds, fraction of sp^3 atoms, and maximum Tanimoto similarity to the training set.*

Our key findings are summarized in Supplementary Table S5. Overall, the correlations are weak, indicating that residual errors are not strongly driven by simple size or flexibility descriptors. The strongest trend observed is a moderate negative correlation ($r \approx -0.2$) between both UniProp error and improvement over Chemprop and the maximum Tanimoto similarity to the training set, suggesting that out-of-distribution chromophores remain the dominant source of error. This trend is consistent across both wavelength and frequency representations.

We additionally observe a weak positive correlation between molecular weight and absolute error when expressed in wavelength units (nm), which disappears in the frequency domain. This behavior is consistent with the known physical relationship between chromophore size and absorption wavelength (larger conjugated systems exhibit red-shifted absorption), and indicates that this effect is largely a unit-dependent artifact rather than a model deficiency.

To provide qualitative insight, we further include visualizations of the ten molecules with the largest UniProp errors and the ten molecules showing the largest improvement of UniProp over Chemprop (Supplementary Figs. S3 and S4). These examples span diverse chromophore families and do not show clear enrichment toward a specific scaffold class, size regime, or flexibility pattern, consistent with the weak descriptor correlations.

Finally, to directly test whether the largest remaining errors could be attributed to insufficient conformational sampling, we performed CREST conformer searches (with GFN-FF potential) for the ten highest-error molecules and evaluated UniProp using the lowest-energy conformer, a Boltzmann-weighted ensemble, and a mean ensemble. As summarized in Supplementary Table S6, none of these ensemble-based evaluations led to a systematic reduction in error compared to the original single-conformer input. This suggests that, for these difficult cases, the dominant error source is unlikely to be an incorrect local minimum from conformer generation.

We now explicitly reference this new Supplementary analysis in the Discussion and clarify that, within the scope of our benchmark, residual errors are primarily associated with out-of-distribution chemistry rather than chromophore size, flexibility, or insufficient conformational sampling.

- In the Discussion, you say "While the cost of generating 3D conformations for UniMol+ is higher compared to 2D models operating on SMILES alone, its superior accuracy far outweighs this additional cost, making it a more reliable choice for applications requiring high predictive fidelity." What is the approximate difference in inference speed (e.g. average per 1000 molecules) for the various 2D and 3D methods? They're all obviously much faster than TD-DFT, and it's great how you show that UniProp can perform well with only the xTB geometry as input at inference time, but it

seems like the extra cost of running xTB (avg 27.54 s/molecule) plus the use of equivariant 3D architectures could still make your best method noticeably slower than the 2D methods. You may be correct that the additional cost could be worthwhile, but I think it's difficult to make a cost-benefit comparison like this without actually quantifying the costs, and quantifying this could help future researchers evaluate this cost-benefit tradeoff for themselves. Additionally, since the 3D methods other than UniProp require the DFT geometries at inference time, providing some timings would further strengthen your case for UniProp as the best choice (though it seems that for most of the other 3D models, using a DFT geometry instead of xTB only reduces MAE by ~1-3nm anyway, and in some cases actually makes it worse when there is no solvent embedding).

Ans: Following your comment, we decided to test the UniProp model in rdkit->dft implicit setting to alleviate the concerns on increased inference time (due to the need to generate xTB conformation). In this setting, the main computational burden is the inference of UniProp model, as generating conformations with ETKDG2 and optimizing with MMFF94 is relatively cheap. The comparison of UniProp trained with different conformation pairs can be found in Section 3.2.

While examining the performance of UniProp models, we found that instead of plotting prediction MAE in figures 4 and 5, we plotted the full training error which is a sum of prediction MAE and position loss. After adjusting the metrics, we found that UniProp performs similarly in all settings (rdkit->xtb being slightly worse, by ~0.3nm). This means that our proposed UniProp model can be directly compared with ChemProp in terms of computational efficiency.

We measured the throughput of UniProp and ChemProp models on Nvidia A100 GPU. The UniProp processes 216.4 samples per second, whereas the ChemProp processes 1764.6 samples per second. This performance gap is due to the size of the UniProp.

- Several places throughout the manuscript, the authors use the word "significantly" but do not perform any statistical significance tests. For example, in the Discussion, "Among the evaluated 3D architectures, UniMol+ and our solvent-aware variant, UniProp, achieved the most accurate predictions, significantly outperforming PaiNN, DimeNet++, GemNet, and eSCN." With no cross-validation comparison for the 3D models and no statistical test, I think it would be more appropriate to use a different word (e.g. "substantial"). The authors also use the word "significantly" several times when comparing 2D to 3D models; although they do cross-validation and report error bars when comparing Chemprop and UniProp, they do not perform any significance test. If the authors would like to make a claim of statistical significance, I suggest they consult this recent paper for best practices: <https://pubs.acs.org/doi/full/10.1021/acs.jcim.5c01609>.

*Ans: We thank the reviewer for pointing this out and agree that the term "significantly" should be reserved for comparisons supported by formal statistical testing. In the current version of the manuscript, we do not perform statistical significance tests across model families for the single held-out test split benchmark, and while we report mean \pm standard deviation across 5-fold scaffold-based cross-validation for Chemprop and UniProp, we did not run an explicit hypothesis test. To avoid overstating the strength of our claims, **we replaced "significantly" with non-statistical wording (e.g., "substantially")** throughout the manuscript, including the Discussion sentence highlighted by the reviewer. We also ensured that any statements comparing models are framed in terms of observed MAE differences under the reported evaluation protocol rather than statistical*

significance. We appreciate the pointer to the referenced JCI article and will follow these best practices in future benchmark extensions where repeated trials / paired tests across folds are performed.

- It appears from the GitHub that the authors have provided the scripts to reproduce the training and inference of their top performing UniProp method. In addition to this, I encourage the authors to also publish (on Zenodo) the trained version of their model so people can easily use it without having to retrain it. It would be even better if it's possible to provide a script to automatically generate the RDKit 3D  xTB conformers and store these in the correct format, so the only thing a user needs to provide to run inference is a list of chromophore SMILES and corresponding solvent SMILES.

Ans: Thank you for the suggestion! We have implemented this in the updated release.

Pretrained UniProp models on Zenodo: we now provide trained UniProp checkpoints on Zenodo (together with the dataset), so users can run inference without retraining: Zenodo record. This includes checkpoints such as RDKit→xTB and RDKit→DFT (implicit solvent).

Workflow for inference directly from SMILES and solvent: we added an example script `04_csv_to_lmdb_rdkit.py` in `examples/conformation_generation/` that takes a CSV with chromophore SMILES and solvent SMILES, generates an RDKit 3D conformer, and writes an LMDB in the UniProp input format. This enables straightforward screening using the `uniprop_rdkit_to_xtb.pt` and `uniprop_rdkit_to_dft_implicit.pt` checkpoints.

xTB / ORCA optimization examples: in the same `examples/conformation_generation/` folder we also added runnable scripts demonstrating how to run xTB geometry optimization from an XYZ (`02_optimize_xtb_from_xyz.py`, including charge and open-shell settings) and ORCA geometry optimization (`03_optimize_orca_from_xyz.py`, with vacuum by default and an optional solvent CPCM line).

#####

Minor questions / suggestions:

- The statement "we filtered 1885 chromophores, either by removing problematic entries or replacing them with corrected structures" is a bit ambiguous, and when I first read it I thought it meant that there were only 1885 chromophores left after filtering out the problematic entries. This is particularly confusing because you don't mention the "13731 individual molecules and 26369 chromophore-solvent pairs" until later in the next paragraph when you're describing the 3D dataset. After inspecting the Zenodo files, I think it could be more clear to say here "we removed 1825 problematic entries and replaced 60 with corrected structures" or something like that, and to state the total number remaining here as well.

Ans: We thank the reviewer for pointing out the ambiguity in this statement and agree that the original wording could be misinterpreted. To improve clarity, we have revised the text to explicitly distinguish between removed and corrected entries and to state the remaining dataset size at this

stage. Specifically, we now clarify that **1825 problematic entries were removed and 60 entries were replaced with corrected molecular structures.**

- I think a brief justification of the choice of DFT functional / basis set for geometry optimizations (r2SCAN-3c/def2-mTZVPP) is warranted in the Methods. The authors mention some justification in the Discussion section ("Specifically, the r2SCAN-3c/def2-mTZVPP level of DFT employed here reliably reproduces interatomic distances and bond angles close to ground truth values").

Ans: We thank the reviewer for this suggestion and agree that an explicit justification of the chosen DFT level for geometry optimization should be stated in the Methods section. We have now added such a justification.

Specifically, we selected the r²SCAN-3c/def2-mTZVPP composite method because it is a well-established best-practice choice for molecular structure optimization, offering an excellent balance between computational cost, robustness, and accuracy. As discussed in the comprehensive best-practice guidelines by Grimme and co-workers [<https://doi.org/10.1002/anie.202205735>], r²SCAN-3c is explicitly recommended for geometry optimizations, as it reliably reproduces interatomic distances and bond angles close to high-level reference data while remaining computationally efficient compared to hybrid or double-hybrid functionals.

We have moved this rationale into the Methods section, clarifying that the choice was motivated by prior large-scale benchmarking and practical experience showing that r²SCAN-3c provides accurate and robust molecular geometries at a fraction of the cost of higher-level DFT methods.

- It would be helpful for you to specify which major version of Chemprop you have used for the study, because the default featurizer and some other things changed between v1.x and v2.x. Additionally, the models trained using v1.x are not compatible with v2.x without first running a conversion script, so I suggest the authors also add this version specification in their GitHub repository where the trained Chemprop model is stored.

*Ans: In this work, we used **Chemprop version 1.3.0**, specifically the version archived at Zenodo (<https://zenodo.org/records/5773155>), which corresponds to the implementation used in the study "Multi-fidelity prediction of molecular optical peaks with deep learning" (DOI: 10.1039/D1SC05677H). We intentionally adopted this version to ensure methodological consistency with prior solvent-aware optical property prediction work.*

We have now explicitly stated the Chemprop version (v1.3.0) in the section S1.4 in the Supplementary materials. In addition, we have updated our GitHub repository to document the Chemprop version used for training and to avoid any ambiguity regarding compatibility with Chemprop v2.x models.

- I suggest the authors emphasize the performance of the TD-DFT B3LYP/def2-TZVPP baseline (62 nm) more, maybe by adding it to the caption of Fig 4 or Fig 5, since this is a key point that all of these methods perform much better than TD-DFT at a fraction of the cost. Normally I might suggest adding a horizontal line for the TD-DFT value for comparison on the bar plots in Fig 4 and 5, but the value of

62nm is so much larger than the ML models that it would distort the rest of the plot, so I think a comparison in the caption could be helpful for someone just skimming the paper.

Ans: We have added the TD-DFT value to the caption of Fig 4.

- I think the captions of Fig. 4 and 5 could mention what level of geometry was used at training and at inference for each of these 3D methods. If I understand correctly, in Fig 4., all of these used DFT with implicit solvent during training, all but UniProp used DFT with implicit solvent during inference, and UniProp used xTB at inference. And in Fig. 5, the different colors of the bars indicate which geometry level is used for training, and the same level is used at inference except in the case of UniMol+ where xTB is used at inference.

Ans: We have updated the captions to include the information about geometries used in training and inference.

- Do I also understand correctly that in Fig. 5, for the right most set of 6 bars, the 3 hatched bars are UniMol+, and the 3 unhatched bars are UniProp, even though you label them collectively as Unimol+ (since UniProp is UniMol+ with the Chemprop solvent embedding)?

Ans: You are correct. We have added separate labels for Unimol+ and UniProp.

- I think the use of 5 significant figures in Table 1 and the following paragraphs is probably overkill (1 or 2 digits after the decimal place should be sufficient for values in nanometers).

Ans: We have reduced the number of digits both in the main text and in the Supplementary.

- When you describe why you dropped some entries from the dataset during cross validation, you say "During cross-validation, we found that UniProp consistently underperformed on one specific split." I suggest mentioning that Chemprop also performed much worse on these same splits, which your results in Table S3 demonstrate. Without saying that, it may sound like the UniProp model is uniquely bad at capturing the solvatochromism of betaine dye36 in many unique solvents.

Ans: We have rewritten the second paragraph of Section 3.3 to better reflect that both ChemProp and UniProp models fail to accurately predict the solvatochromism of betaine dye36 in many unique solvents.

- The authors mention in the Discussion that "In our preliminary experiments, rigorous cleaning not only reduced the MAE but also enhanced training stability, underscoring the importance of high-quality datasets when training expressive 3D GNNs." Are you able to provide any data in the SI to support these claims?

Ans: Thank you for raising this point. Unfortunately, we were unable to retrieve the experimental results that would allow us to quantitatively support this claim in the current manuscript. The statement was therefore based on empirical observations made during dataset construction and early model development, rather than on a systematic ablation study explicitly designed to measure

the impact of data cleaning. To avoid overinterpretation, we have revised the Discussion to remove any implication of a quantitative comparison and now frame this point as a qualitative motivation for the extensive data curation procedure adopted in this work.

- Why did the authors choose to model PLQY in addition to absorption and emission wavelength, but not FWHM bandwidth or other properties that are present in some of the source datasets?

Ans: The main reason why we chose PLQY over other properties is the availability of the experimental data. For example, the FWHM bandwidth data is even more scarce than PLQY data.

- I suggest that the authors specify how the error bars in Table 1 were calculated (standard error?).

Ans: Thank you for the suggestion. We have specified how the error bars were calculated in the caption of Table 1.

- In the abstract and Introduction, the authors mention an improvement of "more than 30%" over the previous best 2D model, but section 2.3 and the Discussion say "nearly 25%".

Ans: Thank you for pointing this out. We have corrected the percentage in Discussion to match the percentage in the Abstract.

- In Table S2, should the "Pretrained" be changed to "Staged" to match the nomenclature described in section S1.3?

Ans: We have changed the column name to "Staged".

- In Table S3, should "Mean Val Loss" be "Mean Test Loss"?

Ans: We call it "Mean Val Loss" as this is the validation loss averaged over folds when performing cross-validation. We reserve the "Test Loss" term for the case when we have both validation and test splits at the same time.

#####

Typos / formatting:

- Inconsistent capitalization of "Chemprop" vs. "ChemProp" throughout, including in the text, tables, captions, and figures. I suggest the authors use the canonical capitalization "Chemprop" (change all "ChemProp" instances to "Chemprop").

Ans: Thank you! We changed all "ChemProp" instances in text to "Chemprop".

#####

Reviewer #2 (Remarks to the Author):

The work in its current form is a benchmark report and a demonstration of model scaling. The manuscript is written mainly for the prediction of optical properties by using 3D GNN models in a solvent-aware manner, combined with the curated dataset and optimized molecular geometries in different quantum mechanical levels as input. Considering that too many model works have been done to improve the prediction accuracy, the manuscript fails to provide more physical insights regarding the absorption and emission properties forecasting, except for the MAE reduction.

I therefore cannot recommend the publication of this manuscript in its current form.

Ans: We respectfully disagree with the assessment. To the best of our knowledge, this work is the first to leverage 3D molecular conformations to improve solvent-aware optical property prediction, enabling pretraining on quantum-chemical quantities directly tied to optical behavior (e.g., the HOMO–LUMO gap). We also show that the accuracy of conformational optimization affects the prediction quality. These observations demonstrate that accurate optical properties prediction depends critically on geometric and electronic-structure information. Consistent with this, UniProp improves upon the prior 2D-GNN state of the art by over 30% (Fig. 4), a gain we argue would be unlikely without incorporating 3D conformations. In addition, we release a carefully controlled benchmark spanning both 2D and 3D GNNs and curate the first quantum-chemical dataset for this task with conformations optimized at multiple quantum chemistry theory levels. In this way, the dataset introduces a new prediction target, on which novel 3D GNN architectures can be systematically tested.